# Bundle recommendation methods considering rating data differences for online retailers

**Yan Fang[1], Qiuqin An[1], Xue Jin[1], Ying Liu[2]\***

1 School of Maritime Economics and Management, Dalian Maritime University, Dalian, Liaoning, China,
2 Anhui Business and Technology College, Hefei, Anhui, China

\* hanta1982@126.com

## Abstract

Bundling has emerged as a pivotal marketing strategy for online retailers, offering mutual benefits to both merchants and consumers in the rapidly expanding e-commerce landscape. Among various types of user behavior data, user-generated product ratings serve as a critical indicator of individual preferences and satisfaction levels. This research proposes a novel bundle recommendation framework that leverages rating disparities to capture nuanced user preferences and unmet demands. To address the challenges of data sparsity and heterogeneity, we develop a two-stage recommendation method. In the first stage, we enhance the completion of sparse rating matrices by integrating collaborative filtering with deep singular value decomposition. A modified cosine similarity function is introduced, incorporating a rating correction coefficient and an item popularity coefficient to improve similarity estimation. In the second stage, we exploit insights from low-rated items to model user dissatisfaction and latent demands. A dual-layer graph self-attention network is constructed to fuse heterogeneous data, refine inter-item relational representations, and enhance bundle recommendation accuracy. Extensive experiments conducted on benchmark Amazon datasets demonstrate the effectiveness of our approach, achieving 3–6% relative improvements in NDCG and Recall metrics compared to state-of-the-art baselines. Moreover, user satisfaction with the recommended bundles also increased significantly. These results highlight the value of rating differences in understanding user behavior and validate the efficacy of our two-stage model in improving bundle recommendation performance for online retailers.

## 1 Introduction

Bundle recommendation has become an essential strategy for online retailers. With the rapid evolution of internet technologies, major e-commerce platforms such as JD.com, Tmall.com, and Amazon.com have expanded rapidly, offering vast product assortments and efficient delivery services to meet the increasingly diverse demands

**Data availability statement:** All relevant data are within the paper and its Supporting Information files.

**Funding:** This work was supported by National Natural Science Foundation of China (71801028).

**Competing interests:** The authors have declared that no competing interests exist.

of consumers. However, these advancements are accompanied by persistent challenges, including information overload and escalating logistics costs, both of which significantly erode profit margins [1]. For instance, JD.com manages over 2.4 million SKUs but averages only 1.3 items per order, while logistics account for 58% of its operating costs per order [2]. As a result, JD.com's net profit margin in the second quarter of 2022 was merely 2.4%. Bundle recommendation serves as a viable strategy to increase the average number of items per order, thereby optimizing logistics efficiency and enhancing overall profitability [3]. In fact, leading retailers such as Amazon and JD.com have already implemented bundle sales to improve operational performance [4]. Thus, developing accurate and personalized bundle recommendation systems is of strategic importance for online retailers.

Despite its practical value, bundle recommendation has received relatively limited attention compared to traditional single-item recommendation systems. Existing recommendation algorithms often underperform in bundle contexts due to three key challenges. First, while single-item recommendation focuses primarily on modeling user–item interactions, bundle recommendation requires capturing the complex interdependencies among multiple items within a bundle. Second, the data sparsity problem, especially in the early stages of user interaction (i.e., cold-start scenarios), hampers the system's ability to make accurate predictions. Third, traditional models generally fail to leverage nuanced user feedback that could guide more personalized bundling strategies.

To better capture user preferences for both individual items and item combinations, recent approaches incorporate user behavioral data such as click logs, browsing history, and purchase records [5,6]. Among these, user-generated ratings offer a more precise and explicit indication of individual satisfaction and preferences. However, the potential of rating differences—i.e., variations in scores assigned by users to different items—remains underexplored. For example, a low rating on a purchased item infers dissatisfaction and implicitly suggest a preference for higher-quality alternatives in the same category.

This research aims to address the limitations of existing methods by leveraging implicit preference signals embedded in rating differences, alongside item-level attributes, to improve bundle recommendation accuracy. To overcome the challenges of rating sparsity, data heterogeneity, and complex item relationship modeling, we propose a two-stage framework. In the first stage, we mitigate sparsity in the rating matrix by combining collaborative filtering with deep singular value decomposition. Additionally, we introduce a modified cosine similarity function that incorporates a rating correction coefficient and an item popularity coefficient to improve similarity estimation. In the second stage, we exploit low-rated item data to infer customer dissatisfaction and hidden preferences. We then construct a dual-layer graph self-attention network that integrates heterogeneous data, enhances the representation of relational features, and uncovers meaningful item associations for bundle generation.

The primary contributions of this research are as follows:

We introduce a rating correction coefficient and an item popularity coefficient, derived from rating disparities, to enhance traditional similarity computations. These additions improve the accuracy of rating prediction for matrix completion.

We propose a bundle recommendation strategy that leverages low-rated data to infer user demand. Our results show that incorporating low-rating information improves both the performance and user satisfaction of recommended bundles.

We design a dual-layer graph attention network that captures inter-item relationships from heterogeneous data sources. This model enhances relational data representation and significantly boosts bundle recommendation effectiveness.

The rest of this paper is organized as follows: Section 2 reviews related literature. Section 3 defines the research problem. Section 4 presents the proposed two-stage recommendation method. Section 5 discusses experimental results and evaluations. Finally, Section 6 concludes the paper and suggests future research directions.

## 2  Literature review

The next section reviews three major recommendation approaches: collaborative filtering, deep learning, and the hybrid methods.

### 2.1  Recommendation based on collaborative filtering

Collaborative filtering (CF) [7] constitutes one of the most prevalent recommendation approaches in current research. The core concept of CF involves identifying users with high behavioral similarity to the target user and subsequently recommending items that these similar users have demonstrated preference for. CF can be further divided into neighborhood-based collaborative filtering (NCF) and model-based collaborative filtering (MCF). Within NCF, user-based and item-based collaborative filtering are common approaches, while MCF includes techniques such as matrix factorization and singular value decomposition (SVD).

The fundamental principle underlying neighborhood-based collaborative filtering centers on similarity calculation between users or items. Scholars have introduced various methods for similarity calculation based on rating data. For example, Ji et al. [8] proposed an algorithm that combines interest similarity with rating differences, enhancing similarity calculation accuracy and recommendation performance. Niu [9] incorporated factors like rating grade, rating authority, and rating recognition to calculate trust between users, integrating these with Pearson similarity through weighting to achieve a more reliable similarity measure than traditional methods. Additionally, Cai et al. [10] introduced a similarity measure based on rating difference and correlation, using it to prefill the rating matrix and significantly improve the recommendation accuracy.

Model-based collaborative filtering (MCF) focuses on building mathematical models to predict user ratings for unknown items. One of the most commonly used methods within MCF is matrix factorization (MF), which decomposes the rating matrix into the product of two smaller matrices. Latent Factor Models (LFM), which leverage this technique, have shown excellent performance in recommendation tasks. Improved models include Bias MF [11], which incorporates bias terms, and Singular Value Decomposition++ (SVD++) [12], which considers neighborhood influence.

User behavior logs can significantly improve the efficacy of product bundle recommendation systems. For example, Moran et al. [13] employed the K-nearest neighbors (KNN) CF method and the Jaccard similarity coefficient to optimize product bundles, taking individualized user preferences into account to facilitate Top-N recommendations. Similarly, Zhang et al. [14] applied web log analysis and user clustering to convert implicit user interests into explicit preferences for items, addressing data sparsity and improving recommendation accuracy. Latent Factor Models (LFM) have undergone continuous refinement in recent research [15]. Notably, Li et al. proposed an innovative commodity bundle coefficient, which mathematically captures item-recommendation factor relationships by extending conventional latent semantic models with collaborative filtering techniques [16].

Collaborative filtering-based recommended systems can be applied to various types of items and bundles. These methods are straightforward to implement and can be enhanced with machine learning to autonomously train matrix decomposition models. However, challenges such as data sparsity [17] and limited generalization capabilities remain.

## 2.2 Recommendation based on deep learning

Researchers have found that applying deep learning techniques to rating matrices can enhance recommendation performance. Deep learning methods enable automatic feature extraction and processing through models, allowing personalized bundle recommendation systems to capture intricate relationships associations between features.

Some researchers have improved collaborative filtering from the encoding perspective. For example, the AutoRec model [18] utilizes an AutoEncoder to reconstruct the co-occurrence matrix and employs custom encoding to predict ratings. Chen et al. [19] further optimized the autoencoder model structure, while Lee et al. [20] proposed the BUIR framework, which replaces negative sampling with two distinct encoding networks and applies random augmentation to the encoder input to alleviate data sparsity issues.

Other researchers have leveraged product images and textual descriptions in their systems using Convolutional Neural Networks (CNN) for recommendations. Zheng et al. [21] developed a deep recommendation algorithm that integrates textual comments to jointly model user behavior and product attributes. Additionally, Defferrard et al. [22], Bronstein et al. [23], and Hamilton et al. [24] have made significant advances using Graph Convolutional Networks (GCN) to address missing items in the rating matrix for bundle recommendations. Monti et al. [25] introduced the first GCN-based recommender approach, training the model with a combined objective function of GCN. Deng et al. [26] enhanced bundle recommendations by extending link prediction to model three-way interactions among customers, products and bundles.

The attention mechanism has also been employed to improve recommendation accuracy. It has demonstrated high effectiveness in machine translation. For example, Chen et al. [27] combined the latent factor model with a neighborhood model that incorporates item- and component-level attention to capture implicit feedback in multimedia recommendations. Han et al. [28] proposed an Adaptive Deep Latent Factor Model (ADLFM) that uses CNN and latent factor models (LFM) to generate item representations, leveraging cosine similarity as an attention factor to determine item weights. Hada et al. [29] suggested that recommendations can be generated through a Plug-and-Play Language Model, with a jointly trained cross-attention network.

## 2.3 Recommendation based on hybrid models

Hybrid methods for recommendation, which combine multiple recommendation techniques, have attracted attention for their ability to address challenges such as inadequate feature learning and the cold start problem. Lv [30] proposed a hybrid personalized recommendation method based on context-aware collaborative filtering and knowledge-based recommendation. This approach leverages a personalized recommendation knowledge model, with experimental results validating its effectiveness. Similarly, Cao et al. [31] introduced a hybrid collaborative filtering method incorporating product attribute modeling and a new recommendation ranking formula that utilizes user reviews, demonstrating superior performance in recommendation tasks.

Numerous studies have shown that, compared to systems relying solely on a single recommendation technique, hybrid methods offer improved recommendation outcomes and higher precision [32–34]. However, it is important to note that the specifics of hybrid recommendation approaches can vary significantly depending on the application contexts. Optimizing hybrid approaches for specific recommendation contexts remains an active research frontier.

## 2.4 Summary

Research on product recommendation for online retailers has gained increasing significance, and a variety of approaches have been explored. Collaborative filtering methods are user-friendly and effective in handling rating data across different recommendation types. Deep learning techniques allow the integration of diverse data sources, while hybrid models further enhance the accuracy of rating predictions. However, when addressing the specific challenges of bundle recommendation with consideration for rating differences, this research focuses on resolving the following critical research questions:

(1) Data Sparsity in Ratings. Compared to other behavioral data, such as click and browsing data, rating data is particularly sparse, resulting in high data sparsity. How to effectively fill these sparse data points is a central focus of this research.

(2) Data Heterogeneity. Data heterogeneity manifests as significant disparities in the structured integer data, unstructured textual data, and product relationship data. This research will focus on integrating multi-source heterogeneous data and deriving actionable knowledge for bundle recommendations.

(3) Inter-Product Relationship Mining. This research aims to conduct an in-depth investigation into the challenges associated with mining inter-product relationships, with the objective of enhancing bundle recommendation systems.

## 3 Problem description

Fig 1 gives an illustration of the research problem. Here an e-tailer offers a range of products. Each product is characterized by both basic attribute data (e.g., name, category, and price) and relational interaction data, including ⟨also_viewed⟩ and ⟨also_bought⟩ associations. Customers generate purchase data by browsing and buying products on the e-commerce platform. For example, user $a$ has purchased three products and assigned a rating of 5 to Item $A$ and 1 to Item $E$, while Item $M$ received no rating. The objective of our research is to recommend personalized product bundles, (Bundle 1 and Bundle 2 in Fig 1), for User $a$, by analyzing heterogeneous data sources, including the product data, purchase history, and rating data with differences.

While existing research usually infers user preferences through rating analysis complemented by interaction data, we argue that rating variance encodes more complex patterns in user satisfaction level and potential demands. For instance, as showed in Fig 1, when User $A$ assigns a low rating to Item $E$, this behavior encodes (1) explicit dissatisfaction with the specific product (Item $E$), and (2) implicit latent demand within the broader product category for alternatives. The goal of

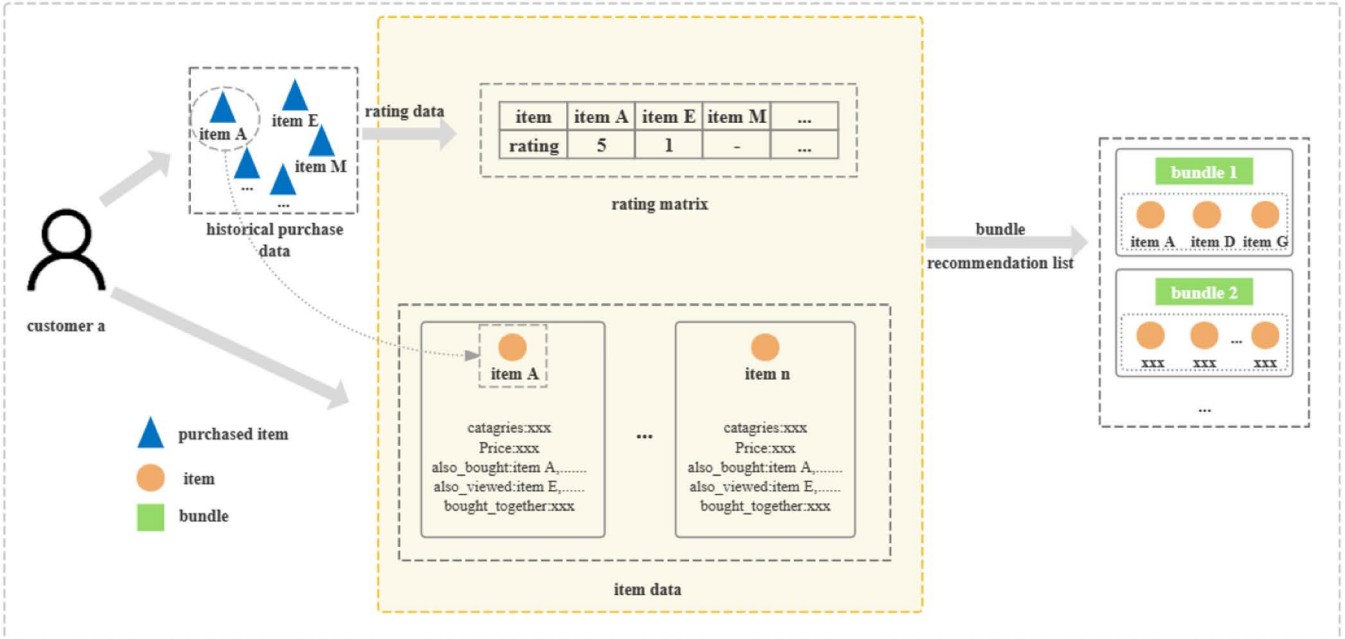

**Fig 1. Problem description diagram.**

this research is to utilize the rating differences to unveil insights into user demands for precise bundle recommendations. Meanwhile, the following challenges remains.

(1) Data sparsity. Taking the dataset of Amazon as an example, the sparsity of the user rating matrix can reach approximately 99%, which leads to issues such as cold start and insufficient available information.

(2) Data heterogeneity. For example, the user rating data and product prices are structured integer data, while product descriptions are unstructured textual data. How to extract valuable information from heterogeneous data is another major challenge.

(3) Requirement of correlation relationships. Unlike single item recommendation, bundle recommendation requires correlations among the items within the bundle. The task to exact correlation relationships from item data, relation data and user rating data, further increases the complexity.

## 4 Methods

### 4.1 Principles of the two-stage method

To address the aforementioned challenges, we introduce a two-stage method for bundle recommendation, as illustrated in Fig 2.

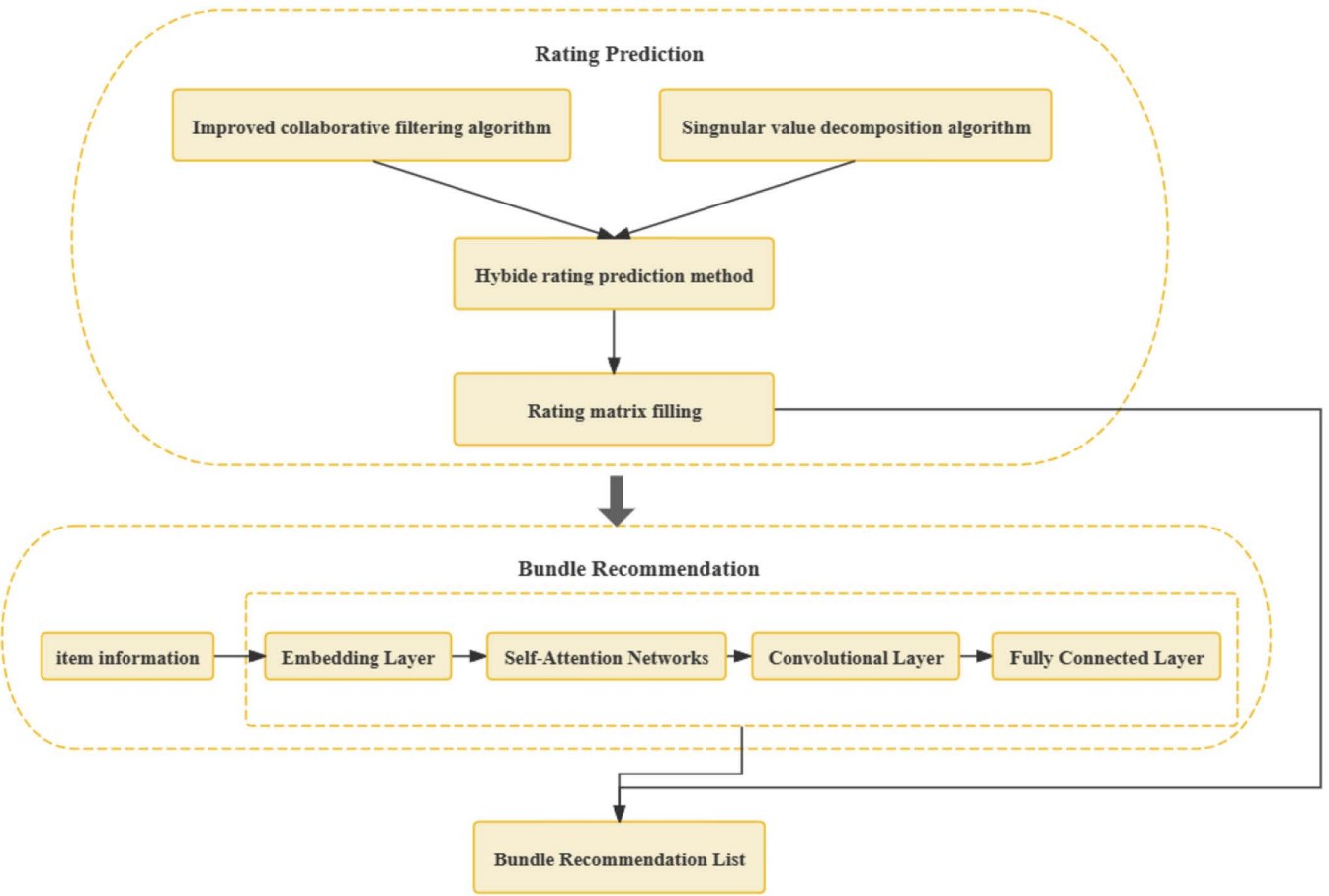

**Fig 2. Illustrative diagram of the two-stage bundle recommendation method.**

In the first stage, we perform the matrix completion via rating prediction to address the sparsity in the rating matrix. We introduce a rating correction coefficient and an item popularity coefficient to adjust similarity calculations, taking into account variations in user rating behavior and item popularity. To enhance model generalization [32], we propose a hybrid approach that adaptively combines the collaborative filtering algorithm and the deep singular value decomposition for rating prediction.

The second stage focuses on bundle recommendation. We construct a dual-layer graph attention network to extract item relationships from multi-source data. First, an embedding layer is created that integrates user data, product data, and product relationship data. We then apply a self-attention mechanism within a dual-layer network to learn optimal weight distributions. Convolutional layers generate node feature pairs, utilizing edge information for relationship prediction among products. Finally, through a fully connected layer, user preferences are combined with the convolutional layer output to generate bundle recommendation.

## 4.2 The hybrid rating prediction methods

**4.2.1 Definitions.** In this subsection, we focus on predicting and imputing the missing entries in the rating matrix to mitigate the problem of data sparsity, commonly known as the *cold-start* issue in recommendation systems. Let $U = \{u_1, u_2, u_3, \ldots, u_{|m|}\}$ and $I = \{i_1, i_2, i_3, \ldots, i_{|n|}\}$ be the user set and the item set. Let $R$ be the rating matrix, where $r_{u,i}$ represents the rating assigned by user $u \in U$ to item $i \in I$, and $r_{u,i} \in R$. The similarity between users is denoted as $sim(\vec{u} \cdot \vec{v})$, where $u, v \in U$.

**4.2.2 The framework.** Considering the limited generalization ability of a single method, we apply the hybrid method combining collaborative filtering with deep singular value decomposition. The framework is illustrated in Fig 3.

The first part is the rating prediction based on the modified similarity-based collaborative filtering. Based on the cosine similarity, we proposed a rating correction coefficient $rev_1$ and an item popularity correction coefficient $rev_2$ to calculate user-to-user similarity. Then we predict the ratings based on the rating data of neighboring users and provides a completion of the rating matrix, denoted as $R1$.

The second model is performed using the deep singular value decomposition approach. The initial rating matrix $R$ is decomposed to reduce dimensionality, resulting in matrices $P$ representing user preference features, and $Q$ representing item attribute features. The original matrix is then filled by training which is denoted as $R2$.

Finally, a linear fusion of $R1$ and $R2$ is conducted based on weighted factors to yield the ultimate rating matrix $R\prime$.

**4.2.3 The models.**

(1) Rating Prediction based on Modified Similarity Collaborative Filtering

The rating data provides two types of valuable information. First, the rating differences can infer the users' preferences and demands. For instance, the higher the rating score is, the higher the preference. Meanwhile, regardless of whether the score is high or low, the rating indicates that there is a demand for items of this category. Second, the popularity of the item can be reflected by how often the item is rated. Based on these insights, we introduce a rating-based correction coefficient and an item popularity correction coefficient to enhance the traditional cosine similarity measure.

1) The Rating Correction Coefficient

The traditional cosine distance calculates the similarity by the angle of two vectors, the traditional cosine similarity equation is shown in Equation (1).

$$sim(\vec{u} \cdot \vec{v}) = \frac{\vec{u} \cdot \vec{v}}{\|\vec{u}\|^2 \cdot \|\vec{v}\|^2} = \frac{\sum_{i \in I_{u,v}} r_{u,i} r_{v,i}}{\sqrt{\sum_{i \in I_u} r_{u,i}^2 \sum_{i \in I_v} r_{v,i}^2}} \tag{1}$$

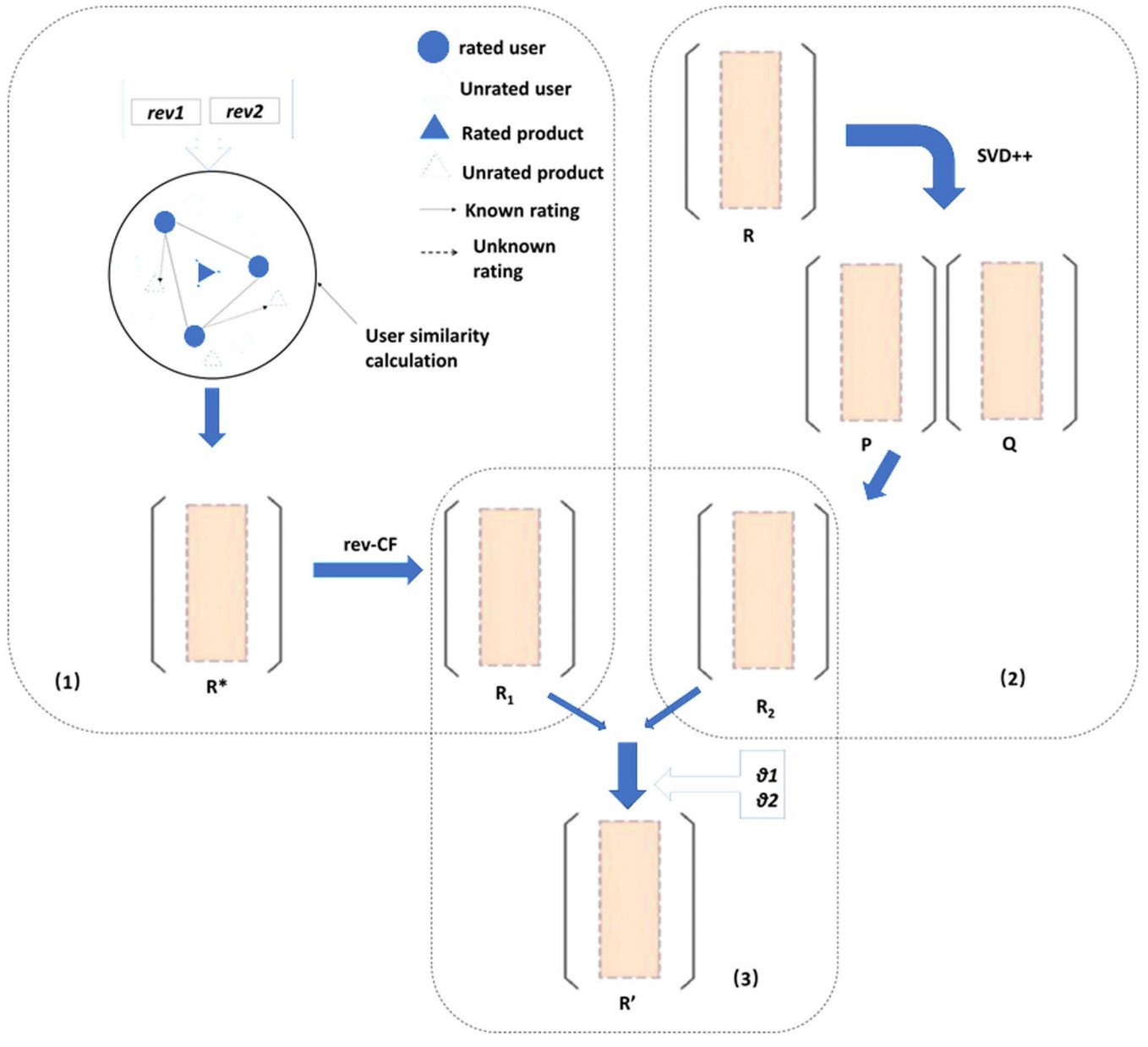

**Fig 3. Framework diagram of the hybrid rating prediction method.**

However, the traditional cosine similarity metric fails to capture the user preference variations embedded in rating differences [35], Therefore, we propose a rating correction coefficient as shown in Equation (2).

$$rev_1\left(\overrightarrow{u}\cdot\overrightarrow{v}\right)=e^{-\frac{1}{n}\sum_{i\in I_{u,v}}\left|r_{u,i}-r_{v,i}\right|}$$

(2)

Here, $rev_1\left(\overrightarrow{u}\cdot\overrightarrow{v}\right)$ represents the correction coefficient that quantifies the difference in rating values between users $u$ and $v$. $n$ denotes the number of items that both users $u$ and $v$ have rated. $r_{u,i}$ and $r_{v,i}$ denote the ratings that users $u$ and $v$

have assigned to item *i* respectively. When the rating discrepancy for a specific item between users increases, the similarity coefficient decreases inversely resulting in a reduction in inter-user similarity.

2) Item popularity correction coefficient

In order to capture the influence of the item popularity on user preferences, we propose the item popularity correction coefficient in Equation (3).

$$rev_2(i) = e^{\frac{N(i)}{|U|} - 1}$$

(3)

Here, $rev_2(i)$ represents the correction coefficient of item *i*. $N(i)$ denotes the overall number of times that item *i* has been rated, and $|U|$ represents the total number of users in the rating matrix. Generally, a popular item will get a higher coefficient value.

3) The modified similarity calculation

By incorporating the two coefficients, we present the modified similarity calculation formula as follows.

$$sim\left(\vec{u} \cdot \vec{v}\right) = rev_1\left(\vec{u} \cdot \vec{v}\right) rev_2(i) \frac{\sum_{i \in I_{u,v}} r_{u,i}\, r_{v,i}}{\sqrt{\sum_{i \in I_u} r_{u,i}^2 \; \sum_{i \in I_v} r_{v,i}^2}}$$

(4)

Where the rightmost fractional component in the equation corresponds to the standard cosine similarity calculation.

4) Rating Prediction Based on the Nearest Neighbor Set

Based on the user's nearest neighbor set, we can infer the predicted rating $\hat{r}_{u,i}^{cf}$ for user *u* to item *i* as follows.

$$\hat{r}_{u,i}^{cf} = \frac{\sum_{v \in N_{i(u)}} sim\left(\vec{u} \cdot \vec{v}\right)\, r_{v,i}}{\sum_{v \in N_{i(u)}} sim\left(\vec{u} \cdot \vec{v}\right)}$$

(5)

Here $N_{i(u)}$ is the top-N nearest neighbor user set to user *u*. $r_{v,i}$ is the actual rating score of user *v* to item *i*, where $v \in N_{i(u)}$ and $sim\left(\vec{u} \cdot \vec{v}\right)$ is the similarity between user *u* and *v*.

The steps for rating prediction based on modified similarity calculation are as follows:

Step 1: Rating Matrix Construction. Construct a rating matrix based on historical user rating information.

Step 2: Correction Coefficients Calculation. Utilize the modified similarity calculation method to compute the similarity between the target user and other users.

Step 3: Neighbor Selection. Sort the similarity results from Step 2 in descending order to obtain the top N nearest neighbor user set.

Step 4: Rating Prediction. Predict ratings for items that the target user has not yet rated.

(2) Rating Prediction Based on the Deep Singular Value Decomposition Algorithm

1) Rating Prediction of Users

The Deep Singular Value Decomposition (SVD++), takes into account the influence of a user's historical behavior on rating prediction. It decomposes the rating matrix, mapping user and item features to a latent factor space with *K* dimensions. The relationships between users and items can be expressed as inner product in this space forming the semantic associations. The fundamental formula [12] for rating prediction based on SVD++ is given in Equation (6).

$$\hat{r}_{u,i}^{svd++} = q_i^\mathsf{T} \cdot \left( p_u + \frac{1}{\sqrt{N(u)}} \sum_{j \in N(u)} y_j \right) \tag{6}$$

Here $\hat{r}_{u,i}^{svd++}$ represents the predicted rating by the SVD++ algorithm for user $u$ on item $i$. The rating matrix is decomposed into $K$ dimensions, where $q_i$ signifies the feature value of item $i$, and $q(i,k)$ denotes the size of the k-th feature for item $i$. Similarly, $p_u$ represents the preference of user $u$, and $p(u,k)$ indicates the preference of user $u$ for the k-th feature. $N(u)$ represents the set of products that user $u$ has purchased and rated. Additionally, $y_j$ represents the feature of items that user has purchased but not rated.

However, there are certain biases in rating. Firstly, customers ratings to a same product varies. Secondly, products may have quality issues, resulting in lower ratings from few users. Additionally, the overall rating on e-commerce platforms or datasets may have systematic differences. For example, all the books may receive lower rating on the platform like Douban.com compared to Amazon.com. The former may have more critical reviewers. Therefore, we applied three bias variables according to [12]: 1) $bu$ is the user bias, representing individual user rating habits; 2) $bi$ is the item bias, accounting for factors in rating that are unrelated to users; 3) $\mu$ is the average rating of all items on the platform or the dataset. The modified user rating prediction formula is as follows:

$$\hat{r}_{u,i}^{svd++} = q_i^\mathsf{T} \cdot \left( p_u + \frac{1}{\sqrt{N(u)}} \sum_{j \in N(u)} y_j \right) + bu + bi + \mu \tag{7}$$

In accordance with the principles of linear regression, the difference between the predicted value and the true value should be systematically minimized. Thus, the loss function for rating prediction is defined as follows:

$$loss = \sum_{u,i \in Train} \left( r_{u,i} - \hat{r}_{u,i}^{svd++} \right)^2 + \lambda \left( \sum_{u \in U} \|p_u\|^2 + \sum_{j \in N(u)} \|y_j\|^2 + \sum_{i \in I} \|q_i\|^2 + \sum_{u \in U} \|b_u\|^2 + \sum_{i \in I} \|b_i\|^2 \right) \tag{8}$$

Here, '$Train$' represents the training dataset, and $r_{u,i}$ represents the observed rating values given by user $u$ to item $i$. We also incorporate a regularization coefficient $\lambda$ into the loss function to prevent overfitting. To minimize the loss function, we employ the Stochastic Gradient Descent Algorithm (SGDA) for training [11].

(3) Fusion of Rating Prediction Models

Finally, we employ a linear weighted fusion to obtain the ultimate rating prediction. The formula for this fusion is given in Equation (9).

$$\hat{r}_{u,i} = \theta_1 \hat{r}_{u,i}^{svd++} + \theta_2 \hat{r}_{u,i}^{cf} \tag{9}$$

Where $\theta_1$ and $\theta_2$ represent the weights of the collaborative filtering algorithm and the deep singular value decomposition algorithm, and $\theta_1 + \theta_2 = 1$. Hyperparameter analysis is conducted to determine the values of $\theta_1$ and $\theta_2$. The primary optimization objective is to minimize the Root Mean Square Error and Mean Absolute Error between the observed rating values $r_{u,i}$ and the predicted rating $\hat{r}_{u,i}$ as shown in eq. (10) and (11).

$$RMSE = \sqrt{\frac{\sum_{u,i \in Test} (\hat{r}_{u,i} - r_{u,i})^2}{|Test|}} \tag{10}$$

$$MAE = \frac{\sum_{u,i \in Test} |\hat{r}_{u,i} - r_{u,i}|}{|Test|}$$

(11)

In Equations (10) and (11) above, *Test* is the test set, and |*Test*| indicates the number of user-item pairs in the test set.

### 4.3 Bundle recommendation based on dual-layer self-attention networks

**4.3.1 Definitions and principles.** Since low-rated items can reveal customer preferences for these products. It is worth considering recommending items from the same category but with higher ratings. Based on this assumption, our recommendation approach is outlined as follows.

Firstly, we define a set of low-rated items for each user, denoted as $I_a$. Given a user set $U = \{u_1, u_2, u_3, \ldots, u_{|m|}\}$, an item set $I = \{i_1, i_2, i_3, \ldots, i_{|n|}\}$, and a satisfaction threshold $r_s$, any item $i$ with a rating $r_{u,i}$ below $r_s$ is included in the unsatisfactory item set $I_a = \{i_{a1}, i_{a2}, i_{a3}, \ldots, i_{|an|}\}$, where $I_a \subseteq I$. The threshold can be adjusted according to the specific application context. For computational convenience, this study adopts the user's average rating $r_{u,i}$, calculated over all observed ratings $\bar{r}_u$.

Secondly, we construct a set of highly related items based on their product category and rating similarity with respect to the unsatisfactory item set $I_a$, denoted as $I_b$. Generally, each item in the list $I_b = \{i_{b1}, i_{b2}, i_{b3}, \ldots, i_{|bn|}\}$ is similar to an item in $I_a$ and belongs to the same category. Then, using the predicted rating matrix $R$, we select $n$ items with higher rating, resulting in the Top-n item list $I_c = \{i_{c1}, i_{c2}, i_{c3}, \ldots, i_{|n|}\}$, which is the set of individual items to be explored for bundles.

Thirdly, we infer the correlated items for items in $I_c$ set. It's evident that items from the same category and similar price levels are more related to target items. Furthermore, the item related data, such as '*also_viewed*' and '*also_bought*', exhibit significant correlations. Therefore, this paper leverages both the basic item data and item relation data as the basic information for bundle exploration. The method for deducing correlated relationship will be described in detail in the following sections.

**4.3.2 Bundle recommendation based on dual-layer graph self-attention networks.**

(1) The model

The relationships among products can be inferred from both the basic item attributes and relation data. The basic data include product ID, price, and category. The relational data comprise '*also_viewed*,' '*also_bought*', '*bought_together*', and '*buy_after_viewing*'. These data sources are heterogeneous in nature, comprising textual, numerical, and relational data. Notably, the relational data directly describe the relationships between items and capture a broader range of correlation patterns. To effectively model these characteristics, we propose a Double-layer Graph Attention Network model (DGAT). The model consists of four main components: an embedding layer, the self-attention network layer, the convolution layer, and the fully connected layer. The overall framework is illustrated in Fig 4.

In the embedding layer, the basic information, relation data and user ratings are represented. In the self-attention layer, employs a dual-layer graph attention mechanism to learn attention weights and enhance the modeling of item relationships. The convolution layer predicts the correlated relationships between items based on edge information. Finally, the fully connected layer recommends bundles based on product correlated relationships and the predicted rating matrix. Details of four layers are illustrated as follows.

1) The Embedding Layer

The input dataset for the embedding layer comprises basic item information and related data.

The basic item data includes product IDs, categories and prices, and they are transformed into node information of the graph. The category of products is unordered discrete textual information. It is initially converted into integers by Label Encoder transformation and converted to Boolean type data using one-hot Encoder [4]. Subsequently, a *z_score*

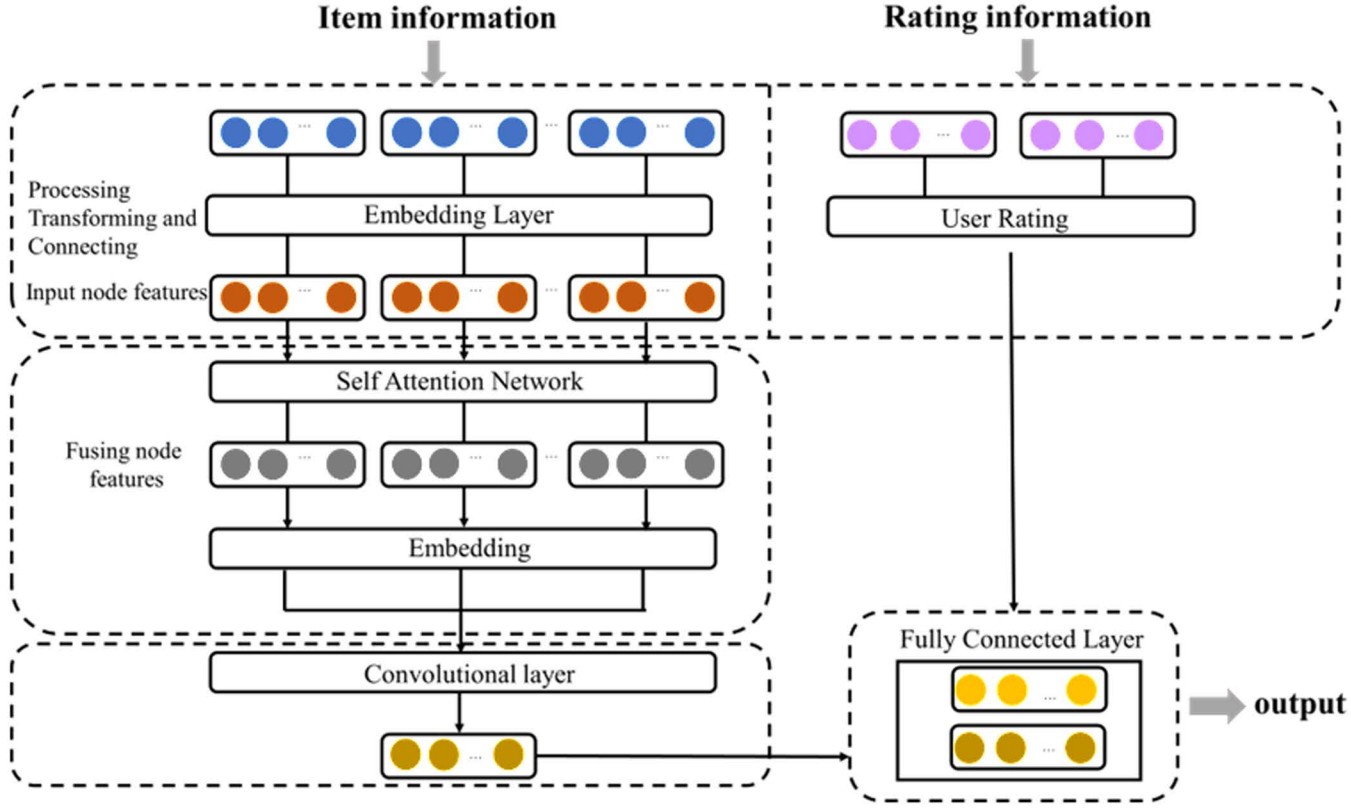

**Fig 4. Framework diagram of bundle recommendation method.**

standardization process will yield a d-dimensional vector. The product prices can be directly standardized using the *z_score*. Finally, these three types of features will be horizontally concatenated, resulting in a node feature matrix h with $d+1 = F$ dimensions, denoted as $h = \left\{ \vec{h_1}, \vec{h_2}, \ldots, \vec{h_N} \right\}$, where $\vec{h_i} \in \mathrm{R}^F$, with $N$ denoting the number of nodes and $F$ representing the number of features within each node.

With respect to item relational data, we first construct an adjacency matrix. If at least one type of relationship exists between two products, the corresponding edge value will be set to 1. The adjacency matrix is defined in Equation (12). Then, the matrix is transformed into edge information within the embedding layer.

$$I = \begin{bmatrix} i^j_{11} & \cdots & i^j_{1|n|} \\ \vdots & \ddots & \vdots \\ i^j_{|n|1} & \cdots & i^j_{|n||n|} \end{bmatrix} \tag{12}$$

Where $i^j_{ij} = 0$ or $i^j_{ij} = 1$.

2) The Graph Self-Attention Networks

The purpose of the attention mechanism is to learn the weights associated with different input features. The Graph Attention Networks (GATs) focus on evaluating the relative importance of nodes by assigning attention weights to the neighbor nodes to quantifying the contribution of each neighbor to the target node's representation. To capture a sufficient

range of node interactions, a double-layer GAT model is constructed, with both layers featuring learnable non-linear transformations. The computation process for the first and second GAT layers is structurally identical, with the input to the second layer being the output of the first. Here we provide an illustrative example of the computational procedure for the first layer of GAT.

We firstly compute the attention coefficients as weights for each target node in relation to its neighbor nodes [36]. In Equation (13), the attention coefficient $e_{ij}$ signifies the importance of neighbor node j with respect to node $i$.

$$e_{ij} = a\left(W\overrightarrow{h_i'}, W\overrightarrow{h_j'}\right)$$

(13)

Here, the matrices $W \in R^{F'} \times R^{F'}$ and $a \in R^{F'} \times R^{F'} \rightarrow R$ are trainable parameters. $\overrightarrow{h_i'}$ and $\overrightarrow{h_j'}$ denote the embedding representations of nodes $i$ and $j$ with the adjustment parameters, $a$ and $W$.

To enhance the comparability of attention coefficients across different nodes, we apply a softmax function to normalize the attention scores for each node $i$. Additionally, LeakyReLU is employed as the activation function to introduce non-linearity and mitigate the dying ReLU problem. The normalized attention coefficients are formally defined in Equation (14).

$$\alpha_{ij} = softmax_j\left(e_{ij}\right) = \frac{exp\left(e_{ij}\right)}{\sum_{k \in N_i} exp\left(e_{ij}\right)}$$
$$= \frac{exp\left(LeakyReLU\left(\overrightarrow{a}^T\left[W\overrightarrow{h_i'} \| W\overrightarrow{h_i'}\right]\right)\right)}{\sum_{k \in N_i} exp\left(LeakyReLU\left(\overrightarrow{a}^T\left[W\overrightarrow{h_i'} \| W\overrightarrow{h_i'}\right]\right)\right)} \#$$

(14)

Here, the superscript.$^T$ denotes the transpose operation, and $\|$ represents the concatenation operation. The normalized attention coefficients are utilized to compute the linear combinations corresponding to their respective features, serving as the final output features for each node.

The computation process in the second layer is similar with the first layer. In this layer, after the self-attention calculation, we can get the $F'$ feature representations for N nodes as $h' = \left\{ \overrightarrow{h_1'}, \overrightarrow{h_2'}, \dots, \overrightarrow{h_N'} \right\}$, where $\overrightarrow{h_i'} \in R^{F'}$. The calculation of $\overrightarrow{h_i'}$ is given in Equation (15), where $\sigma$ is the potential non-linear transformation representation. After the second layer self-attention calculation, we can get a now adjacency matrix $I'$

$$\overrightarrow{h_i'} = \sigma\left(\sum_{j \in N_i} \alpha_{ij} W \overrightarrow{h_i'}\right)$$

(15)

Fig 5 illustrates the non-linear computation process within the self-attention network. It can be observed that, following the double-layer attention computations, the relational representations between products are significantly enhanced. For instance, through double-layer iterations, it can be deduced that there may exist a certain degree of correlation relationship between nodes $h_1$ and $h_2$, which were initially unconnected.

3) The Convolutional Layer

We model the problem of inferring correlations as an edge prediction training task based on Graph Neural Networks (GNN). For example, after self- attention network, we can get the features of each node. Given node $A$, as well as its related nodes $B$, $C$ and $D$. Similar positive sample pairs are $(A, B)$, $(A, C)$, and $(A, D)$. By computing the dot product of d-dimensional node features for $(A, B)$, $(A, C)$, and $(A, D)$ separately, we can get the values to predict the correlation degree for pairs. A higher value represents a higher correlation degree.

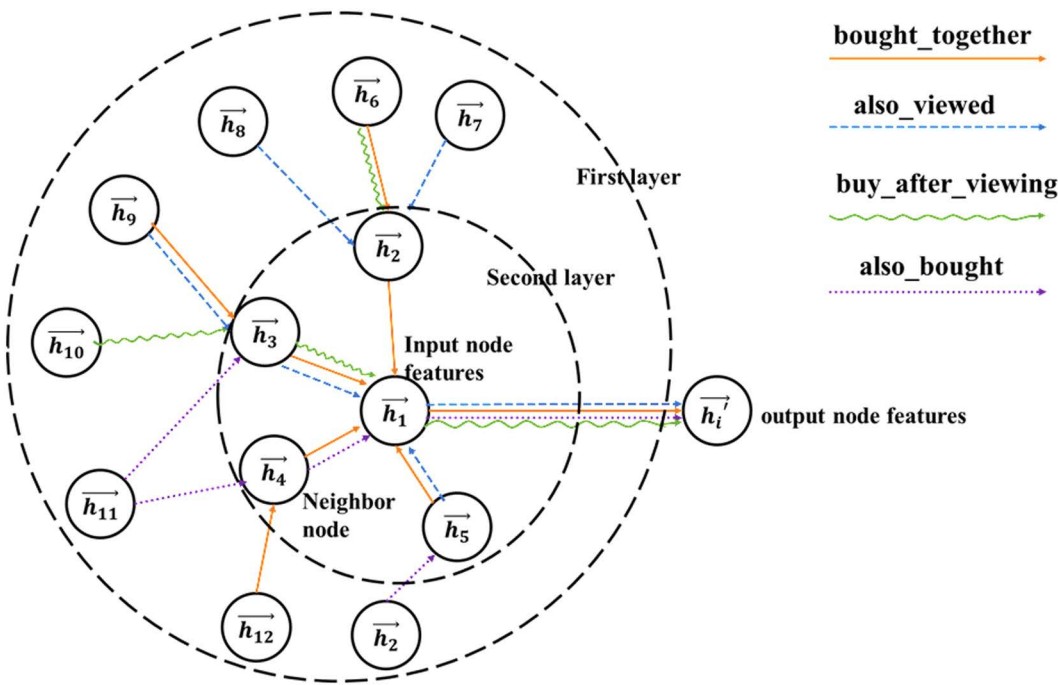

**Fig 5. Calculation process diagram of the attention layer in GAT.**

4) The Fully Connected Layer

We concatenate the predicted rating matrix and the output of the convolutional layer, which respectively represent user preferences and item relational features. This combined representation is then fed into a fully connected neural network to predict bundles with high user interest. Finally, the Top-n bundles are selected as the recommendation results.

(2) Model Training

This paper adopts a Negative Sampling strategy to train the model. Specifically, for each target node, five nodes that are not connected to it in the graph structure are randomly selected to form negative sample pairs. Each sample pair is then processed using a dot product operation to produce a prediction value $\hat{a}$. Ideally, the prediction value $\hat{a}$ for negative samples should be close to 0, while that for positive samples should approach 1. The better the model, the closer the predictions align with this pattern.

Since the Hinge Loss function is commonly used in classification tasks [37], it is employed here to reflect the classification of edges in the graph. Accordingly, the loss function used in our model to capture the association strength between products is defined as follows equation (16), where $a$ denotes the true label, taking the value $+1$ for positive samples and $-1$ for negative samples.

$$loss(a) = \max\left(0, 1 - a \cdot \hat{a}\right) \tag{16}$$

## 5 Experiments

### 5.1 Datasets

In this experiment, we employed publicly available datasets from Amazon, and three widely used sub-datasets are selected [38], namely *Toys_and_Games*, *Beauty*, and *Movies_and_TV*. We filtered out users with less than 20 reviews and items with less than 20 ratings. The summary dataset information is presented in Table 1 below.

The calculation formula for sparsity in Table 1, which describing the sparsity level of item ratings [39,40], The formula is as follows. It reflects how full the user–item rating matrix is compared to its maximum possible size. In an ideal situation, each user rates each item, in fact, only a small part of items is actually rated.

$$\text{Sparsity} = 1 - \frac{N_{ratings}}{|U| \times |I|} \tag{17}$$

Where $N_{ratings}$ denotes the number of ratings, while $|U|$ and $|I|$ represent the total number of users and items respectively.

## 5.2 Results and analysis of rating prediction

**5.2.1 Settings.** In order to identify the optimal parameter settings for the model, we performed a series of experiments across different hyperparameters. In the rating prediction experiments, the number of latent features, $F$, were tested within the range of $\{5, 15, 25, 35, 45\}$. The gradient descent step size, was experimented with values from $\{10, 20, 30, 40\}$. The regularization coefficient, $\lambda$, was set to 0.1; the learning rate, $\alpha$, was fixed at 0.05.

Additionally, given the sparsity of user-item interaction data in our study, we set the number of neighbors to 50 to balance between overfitting and underfitting [12,41], and the batch size was set to 256.

**5.2.2 Effectiveness of rating prediction method.** The proposed rating prediction method is compared with the following four benchmarks, namely the Collaborative Filtering (CF) [42], the Latent Factor Models (LFM) [12], the Singular Value Decomposition (SVD) and the Double Topics with Matrix Factorization (DTMF) [43]. The comparison results are generated based on the metric of RMSE and shown in Table 2.

Table 2 demonstrates that the proposed hybrid method achieves superior performance relative to the baseline models, achieving up to a 13.61% improvement over the best-performing benchmark, DTMF. The method demonstrates its strongest performance on the Beauty dataset. This can be attributed to the high user-to-item ratio in this dataset, which implies that items are rated more frequently and thus offer richer data for neighborhood-based calculations. Consequently, greater availability of behavioral data are available for rating prediction. Notably, the Movies_and_TV dataset exhibits the highest overall improvement among the datasets tested. Directly validates its extreme sparsity, which highlights the robustness and effectiveness of the proposed data completion strategy under sparse conditions.

**Table 1. Summary of experimental dataset.**

|  | Toys_and_Games | Beauty | Movies_and_TV |
|---|---|---|---|
| **Number of users** | 413 | 2826 | 9821 |
| **Number of rated items** | 3142 | 835 | 16140 |
| **Number of ratings** | 16455 | 26881 | 590308 |
| **Sparsity(%)** | 98.96 | 98.86 | 99.63 |

**Table 2. The RMSE results of each rating methods.**

| Methods / Datasets | CF | LFM | SVD | DTMF | Ours | Increasement(%) |
|---|---|---|---|---|---|---|
| **Toys_and_Games** | 1.2457 | 1.2462 | 1.1208 | 1.0416 | 0.9274 | 10.96 |
| **Beauty** | 1.2394 | 1.2405 | 1.1127 | 1.0313 | 0.9245 | 10.36 |
| **Movies_and_TV** | 1.2579 | 1.2588 | 1.1283 | 1.0805 | 0.9334 | 13.61 |

**5.2.3 Effective analysis of the revised coefficient.** We conducted systematic ablation studies to quantify the individual contributions of rating correction coefficient and item popularity correction coefficient. We designed two variants: $w/o\ rev1$ indicates the removal of correction coefficient, while $w/o\ rev2$ signifies the removal of item popularity correction coefficient. $rev-CF$ represents a modified rating prediction model. The experimental results, as depicted in Fig 6, reveal that $rev-CF$ outperforms $w/o\ rev1$ and $w/o\ rev2$, and it can demonstrate the effectiveness of the correction coefficient proposed in this study.

**5.2.4 Hyperparametric analysis.**

(1)   Analysis on the latent factors $F$ and the gradient descent step size $N$

The primary parameters for the deep singular value decomposition method include the number of latent factors, $F$, the gradient descent step size, $N$, the regularization parameter, $\lambda$, and the learning rate $\alpha$. In this paper, we use stochastic gradient descent to investigate the influence of $F$ and $N$, with fixing $\lambda = 0.1$ and $\alpha = 0.05$. By changing the values of $F$ and $N$, we observe the RMSE and MAE across three datasets to analyze the impact of these parameters on the results. The experimental results are presented in Fig 7, and the following conclusions can be drawn.

The changes in RMSE and MAE are generally consistent across the three datasets. The optimal gradient descent step size (Step N) is 30, and the optimal number of features ($F$) is 35. Generally speaking, the results fluctuate with the increase of $N$. Taking Fig 7(a) as an example, when the iteration step size, $N$ is between 10 and 20, the RMSE increases, and it forms a concave shape between 20 and 40. The results also exhibit different patterns with the change of $F$. Overall, when $F$ equals to 5, 25, and 45, the changes are relatively stable, with the RMSE steadily increasing as $N$ increases. However, when $F$ equals to 15 and 35, the RMSE shows greater fluctuations.

The results indicate that a proper iteration step size requires calibration within bounded range. Suboptimal convergence rates were observed when employing undersized step sizes, whereas excessive magnitudes induced overly rapid iterations which will miss the potential optimal solution. Additionally, the variation in the number of features, $F$, has a noticeable impact on the RMSE values. Specifically, insufficient few features for training which may lead to underfitting. Conversely, when $F$ exceeds a certain value, training with an excessive number of features may result in the inclusion of ineffective features. This not only fails to improve prediction accuracy but also increases time and space costs.

(2)  Analysis on the parameters of $\theta_1$ and $\theta_2$

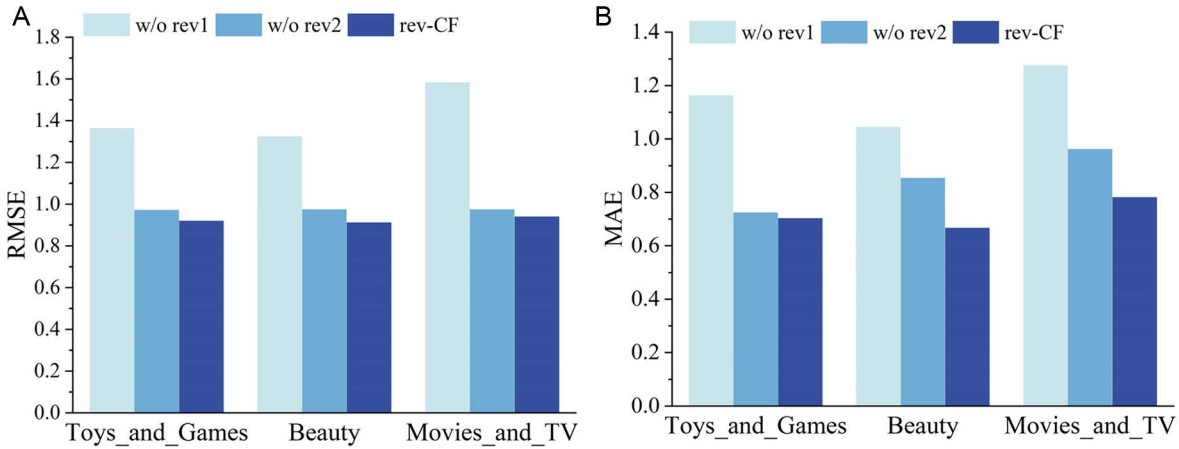

**Fig 6. Performance of rev-CF after removing different correction factors.**

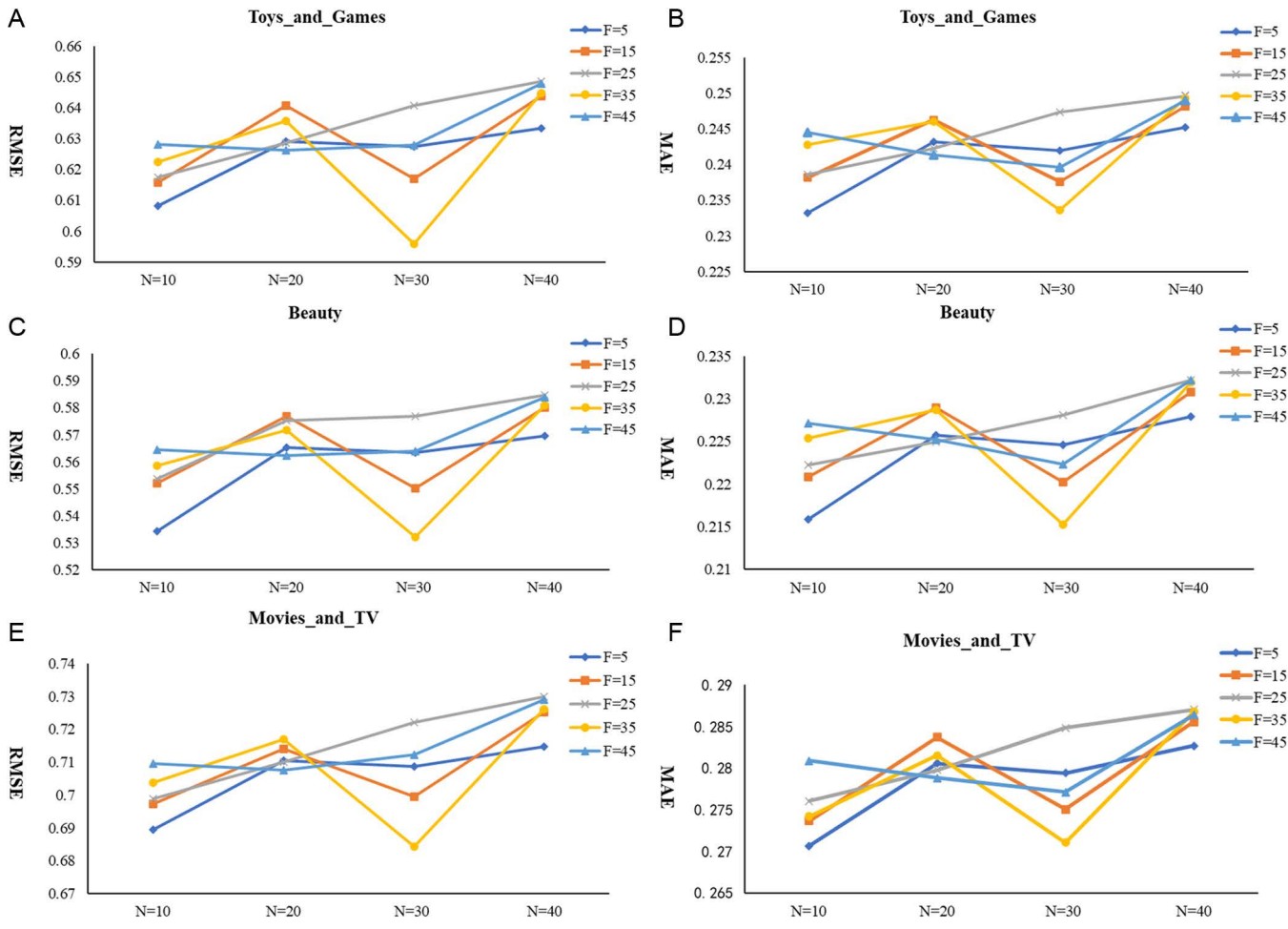

**Fig 7. RMSE and MAE values of the SVD++ algorithm for various F and N values on three datasets.**

The parameters of $\theta_1$ and $\theta_2$ demonstrate the weight of the hybrid rating prediction method defined in Equation (9). When determining the optimal values $\theta_1$ was set from 0.01 to 0.99 with an increment of 0.01, while $\theta_2$ is equals to $(1 - \theta_1)$. The experimental results are shown in Fig 8, where the x-axis represents the Ratio of $\theta_1$ values, and the y-axis represents the RMSE and MAE values of the merged rating prediction methods for each dataset.

It can be observed that the graphs exhibit concave characteristics, suggesting the presence of an optimal $\theta_1$ that yields the best prediction results. However, the optimal point may vary slightly with the evaluation metric and dataset. To address this issue, we determine the optimal $\theta_1$ by summing the RMSE and MAE values when setting the final training parameters, as shown in Fig 9. The results from all three datasets were averaged to obtain the final parameter values for the hybrid rating prediction.

### 5.3 Results and analysis of bundle recommendation

**5.3.1 Settings.** Considering the computational resource constraints, we selected the dataset of *Toys_and_Games* to validate the effectiveness of the bundle recommendation model. The dataset was randomly divided with an 80:20 train-test split ratio. We evaluated the effectiveness of the bundle recommendation using two metrics: Normalized Discounted

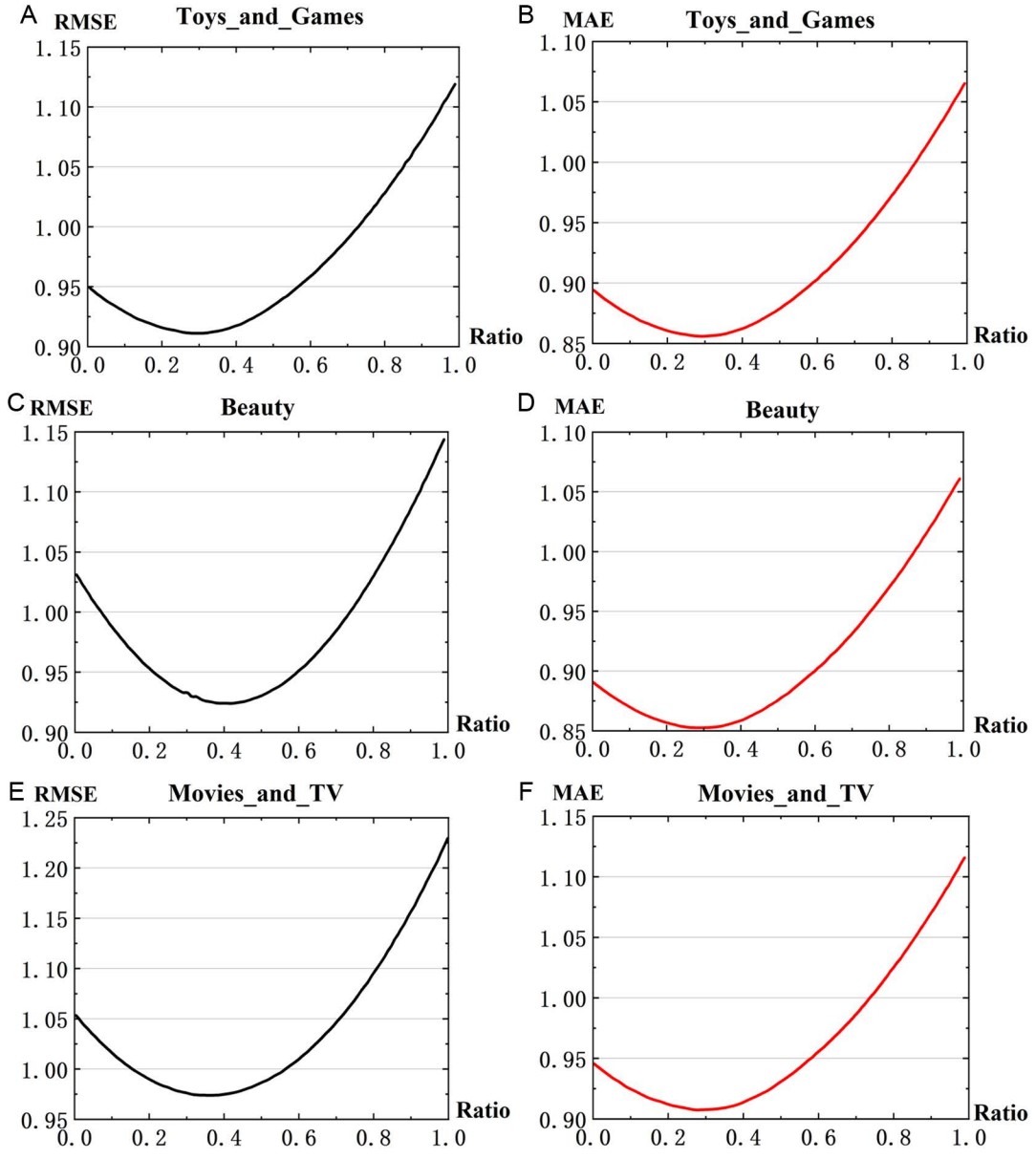

**Fig 8. Impacts of $\theta_1$ on rating prediction RMSE and MAE.**

Cumulative Gain (*NDCG@N*) and Recall (*Recall@N*), where the recommendation list length parameter *N* was evaluated at cutoffs of {20, 50}.

Through a series of experiments and selection, our parameter settings are as follows. The batch size is set to 256. The item feature dimension $d$ is selected from $\{64,\ 128,\ 256,\ 512\}$. The number of self-attention layers is chosen from $\{1,\ 2,\ 3,\ 4\}$. Activation functions are selected from {*rule*, *tanh*, *sigmoid*}. The attention mechanism dimension is set to 400. The number of items in a bundle $M = 2$. And the epoch is set to 300.

**5.3.2 Effectiveness of bundle recommendation.** The baselines are selected from two categories, namely the individual-item recommendation, and bundle recommendations. The first one includes NARRE (Neural Attentional

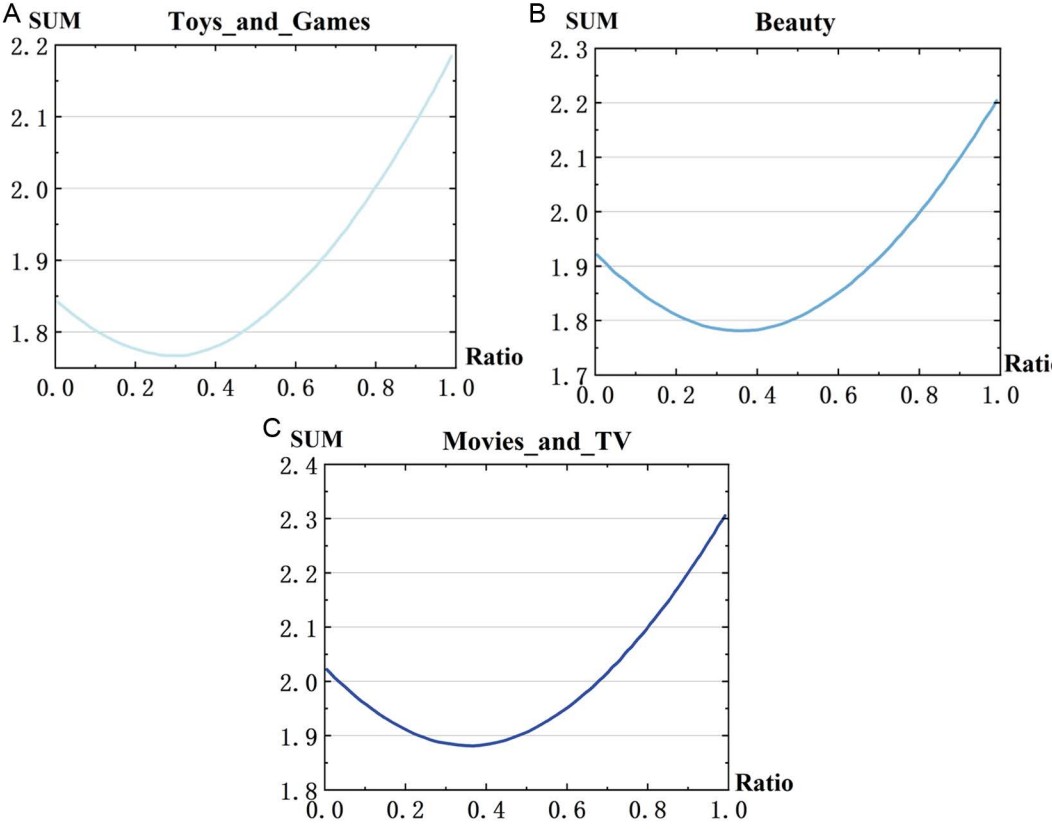

**Fig 9. Impacts of $\theta_1$ on rating prediction (RMSE+MAE).**

Regression model with Review-level Explanations) [44] and HRDR (Hybrid neural Recommendation model learn Deep Representation) [45]. The second category consists of three methods, namely BPR (Bayesian Personalized Ranking) [46], BBPR (Bundle Bayesian Personalized Ranking) [47], and SUGER (Subgraph-based Graph Neural Network) [48].

We report the average metric values in Table 3. The results clearly indicate that the proposed method outperforms all baseline approaches. By leveraging low-rated items to infer user dissatisfaction and latent preferences, our method achieves more accurate predictions of user-preferred products compared to traditional techniques. Notably, the proposed model, along with SUGER and BBPR, consistently surpasses the other three baselines. This performance advantage can be attributed to their shared ability to incorporate product relationship data into the recommendation

**Table 3. Results of bundle recommendation for each method.**

|  | Commodity recommendation method | | Bundle recommendation method | | | Ours | Increment (%) |
|---|---|---|---|---|---|---|---|
|  | NARRE | HRDR | BPR | BBPR | SUGER |  |  |
| **NDCG@20** | 0.3381 | 0.3576 | 0.3221 | 0.3764 | 0.3802 | 0.3942 | 3.55 |
| **NDCG@50** | 0.4369 | 0.4852 | 0.4645 | 0.5203 | 0.5193 | 0.5397 | 3.78 |
| **Recall@20** | 0.2415 | 0.2561 | 0.2637 | 0.2713 | 0.2796 | 0.2988 | 6.43 |
| **Recall@50** | 0.2981 | 0.3234 | 0.3453 | 0.3613 | 0.3922 | 0.3947 | 0.63 |

process. Furthermore, our model employs a two-layer attention-based network architecture for nonlinear feature learning, which enables the extraction of more complex inter-item relationship features and leads to more effective bundle recommendations.

**5.3.3 Effectiveness of bundle recommendation considering low-rating data.** We conducted the ablation experiment to validate whether considering low-rating data can improve recommendation performance. We designed a variant by removing the satisfaction coefficient associated with low-rating data. Additionally, we defined a satisfaction metric to reflect users' satisfaction level with the recommendation results. Its calculation is shown in Equation (18), which is the average ratio of the average ratings of recommended bundles to the target user to the user's historical average rating. A higher value means a higher satisfaction level on the recommended bundles and the maximum value is 1.

$$Satisfcation\ level = \frac{1}{n}\sum_{u=1}^{n}\frac{\bar{r}_t}{\bar{\bar{r}}_u} \tag{18}$$

$$\bar{r}_t = \frac{1}{m}\sum_{t=1}^{m}\bar{r}_{it} \quad,\quad i \in t \tag{19}$$

Where $t$ represent the recommendation list, which is the set of all recommended items. $\bar{r}_t$ represent the average ratings of the bundles recommended to the target user i and $\bar{r}_u$ represent the user's historical average rating.

From Table 4, it can be observed that considering low-rating data induces systematic improvements. When $N$ is set to 20, the NDCG improves by 3.25%, the Recall by 5.46%, and the satisfaction by 10.17%. When N is set to 50, NDCG improves by 4.34%, Recall by 3.07%, and satisfaction by 11.11%. It is evident that the method considering rating differences enhances prediction accuracy and user satisfaction.

**5.3.4 Hyperparameter analysis.** We conducted sensitivity analysis on key parameters in the bundle recommendation model, including the node embedding dimension $d$ on GAT, the number of GAT layers, and the size of bundles.

(1)  Impacts of the node embedding dimension of GAT

Fig 10 illustrates the influence of the node embedding dimension of GAT, $d$, on the results of bundle recommendation. In this analysis, $d$ is varied within the range $\{64,\ 128,\ 256,\ 512,\ 1028\}$.

From Fig 10, it can be observed that the model's performance increases as $d$ is set below 256. However, when the embedding dimension exceeds 256, there is no significant improvement but even a marginal performance degradation. This might be attributed to overfitting caused by the excessively large embedding dimension. Consequently, in subsequent experiments, we set the node embedding dimension of GAT to 256.

(2)  The impact of the number of attention layers in GAT

**Table 4. Results comparison in the ablation experiments.**

|  | Variant | Regular | improvement(%) |
|---|---|---|---|
| **NDCG@20** | 0.3814 | 0.3942 | 3.25 |
| **NDCG@50** | 0.5163 | 0.5397 | 4.34 |
| **Recall@20** | 0.2825 | 0.2988 | 5.46 |
| **Recall@50** | 0.3793 | 0.3913 | 3.07 |
| **Satisfaction@20** | 0.6436 | 0.7164 | 10.17 |
| **Satisfaction@50** | 0.6734 | 0.7576 | 11.11 |

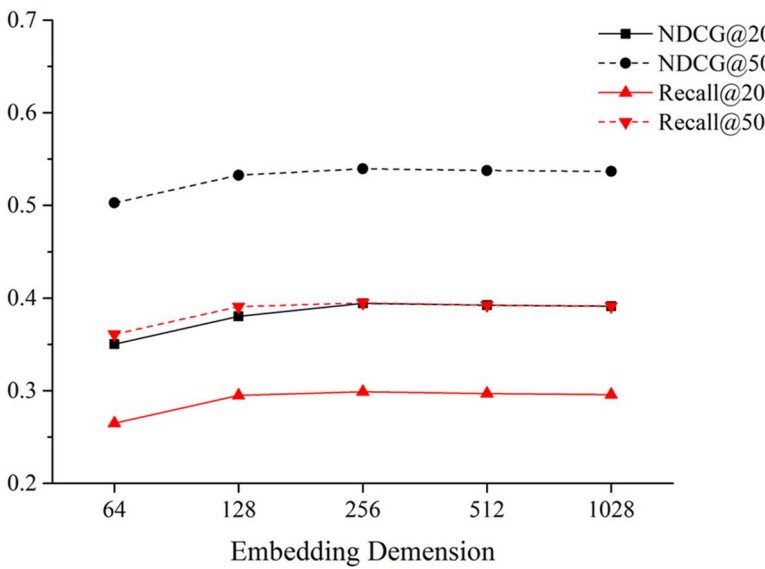

**Fig 10. Impacts of embedding dimension on the model.**

Fig 11 illustrates the impact of the number of attention layers in GAT with its value is ranging from 1 to 4. From Fig 11, it can be observed that when the number of GAT layers is set to 2, the proposed method in this paper has already achieved the best results, and increasing the number of layers does not improve the model's performance. When the number of GAT layers is set to 4, the model's performance metrics start to decline. This could be due to over-smoothing, as aggregating information from multiple layers of neighboring nodes leads to similar information representations for arbitrary nodes.

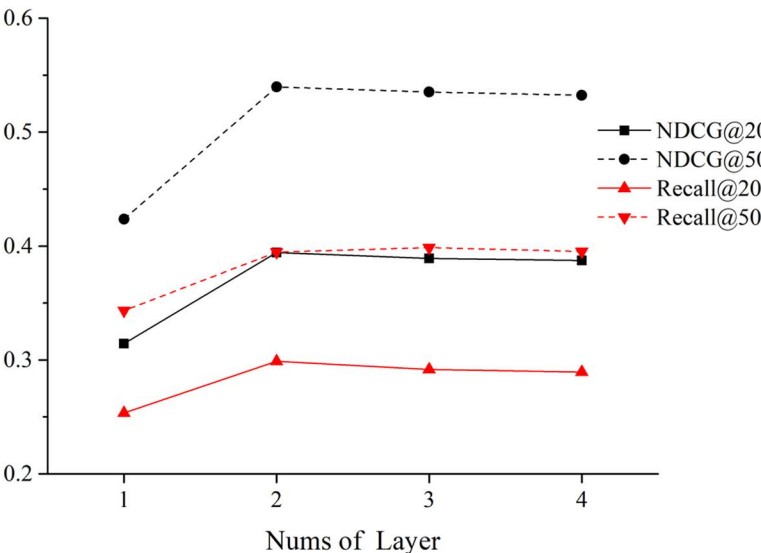

**Fig 11. Impacts of the number of layers on the model.**

**(3) The impact of the size of the bundle**

We set the size of bundle, *L*, from 2 to 7 with an increasement of 1. Fig 12 illustrate the changes in NDCG and Recall values as the size of the bundle varies. It can be observed that when size is set to 6, the recommendation results are optimal. Further increasing the value of size does not lead to improvement in the results. The reason might be that as the number of products in the bundle increases, the predicted ratings for recommended products gradually decrease, and the bundle relationships between products weaken, resulting in a decrease in recommendation effectiveness.

**5.4 Discussions**

This study leverages rating differences to enhance recommendation accuracy and generate more coherent and meaning-ful product bundles. Experimental results demonstrate that the proposed hybrid data-filling method significantly outper-forms existing baselines, achieving up to a 13.61% improvement over DTMF, even under high data sparsity conditions. The effectiveness of the proposed rating correction coefficient and item popularity coefficient has also been validated. These enhancements allow the model to more accurately capture user preferences by accounting for both rating variance and item popularity.

Furthermore, the proposed method exhibits strong performance on multiple Amazon datasets, highlighting the limita-tions of relying solely on neighborhood-based approaches. By integrating collaborative filtering—capable of capturing explicit user interests—with the matrix decomposition capabilities of SVD++, the model uncovers latent preferences informed by historical user behavior. This hybrid strategy enables a more holistic understanding of user interests, thereby improving the accuracy of rating predictions.

In addition, our model surpasses established baselines [4,43–46], confirming that incorporating low-rated item data can reveal valuable insights into user dissatisfaction and unmet needs. This leads to more precise predictions of user-preferred items and bundles. The architecture's flexibility also allows it to be extended to other recommendation domains—such as movies or music—by applying targeted transformations to user and item data.

The dual-layer self-attention network plays a pivotal role in enhancing bundle recommendations by extracting deeper and more nuanced inter-item relationships through nonlinear learning. However, the choice of network depth must be

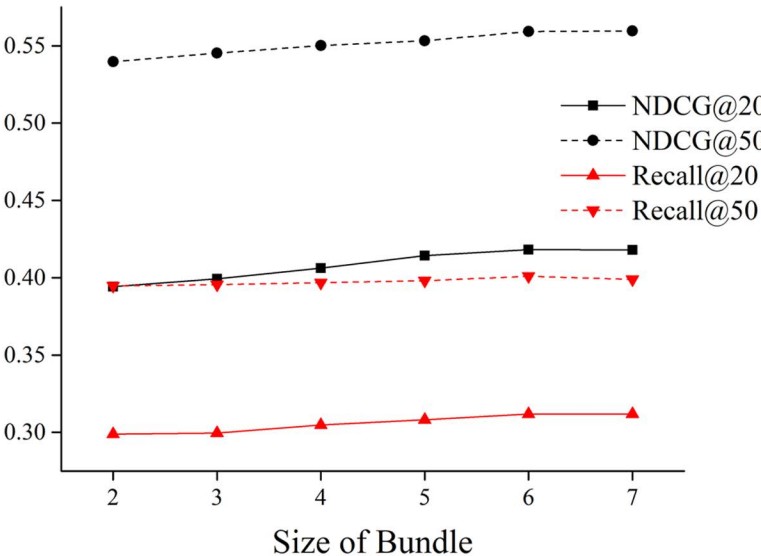

**Fig 12. The impact of the bundle size on the model.**

approached cautiously. Increasing the number of layers does not always lead to better performance and may instead cause over-smoothing, where excessive aggregation of neighbor information results in indistinct node representations and reduced model effectiveness.

Despite the promising results, some limitations remain. One key concern is the presence of unreliable data in real-world environments, such as maliciously low or artificially inflated ratings, which can distort user profiles and compromise recommendation accuracy. Additionally, while the model provides effective bundle predictions, it currently lacks explainability features. Without transparent reasoning behind recommendations, user trust and acceptance may be diminished.

## 6 Conclusions

This paper addresses the challenge of product bundle recommendations in electronic commerce, particularly concerning data sparsity and heterogeneity. We propose a novel two-stage method to tackle these issues. In the first stage, we fill the sparse rating matrix using a hybrid approach that combines collaborative recommendation with deep singular value decomposition. This method introduces rating correction and product popularity coefficients to enhance the traditional cosine similarity function, resulting in improved rating prediction accuracy.

In the second stage, we leverage low-rated data to mine customer demand and preferences, integrating satisfaction coefficients for more effective bundle recommendations. Additionally, we present a product association calculation model based on a dual-layer attention network, which enhances the representation of relational data and improves overall recommendation performance. Unlike existing literature, our approach fully utilizes the differences in user rating values to derive insights into customer demand and preferences.

The key contributions of this research are summarized as follows:

(1) We propose a rating correction coefficient and a product popularity correction coefficient based on differences in rating values, which enhance the traditional cosine similarity function. The results demonstrate that these coefficients effectively improve rating prediction accuracy.

(2) We introduce an approach to product bundle recommendations that leverages low-rated data, fully utilizing customer demand as reflected in low ratings and combining satisfaction coefficients for enhanced recommendations. Our findings show that recommendations based on low-rated data not only improve overall recommendation metrics but also increase customer satisfaction with the recommended products.

(3) By focusing on product attributes and inter-product relationships, we present a product association calculation model based on a dual-layer attention network. This model effectively integrates heterogeneous data, enhances the representation of relational data, and ultimately improves overall recommendation performance.

Our experiments validate the effectiveness of the proposed method, achieving approximately 3–6% relative improvements over state-of-the-art models in terms of NDCG and Recall. Furthermore, user satisfaction levels also saw notable improvement. These results underscore the significance of rating data in bundle recommendations and highlight the efficacy of two-stage models in enhancing recommendation performance for online retailers.

Future research should explore mechanisms to detect and filter out anomalous rating behaviors, as well as develop explainable recommendation frameworks that enhance transparency and interpretability. Moreover, incorporating adaptive mechanisms that dynamically adjust the weighting of rating differences and inter-item relationships could further improve model robustness and adaptability to evolving user behaviors and data patterns.

## Supporting information

**S1 Text.   S1 File. The main code of the research.** S2 File. Data generated during research.
(ZIP)

## Author contributions

**Conceptualization:** Yan Fang, Ying Liu.

**Formal analysis:** Yan Fang, Ying Liu.

**Methodology:** Xue Jin.

**Software:** Xue Jin, Qiuqin An.

**Visualization:** Qiuqin An, Xue Jin.

**Writing – original draft:** Yan Fang, Qiuqin An, Xue Jin, Ying Liu.

**Writing – review & editing:** Yan Fang, Qiuqin An.

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
