## [Decision Letter · Decision Letter 0]

26 Mar 2025

PONE-D-24-56685Bundle recommendation methods considering rating data differences for online retailersPLOS ONE

Dear Dr. Liu,

Thank you for submitting your manuscript to PLOS ONE. After careful consideration, we feel that it has merit but does not fully meet PLOS ONE’s publication criteria as it currently stands. Therefore, we invite you to submit a revised version of the manuscript that addresses the points raised during the review process.

Please see the reviewer comments below. There are some methodological issues raised that should be addressed.  Please note that we have only been able to secure a single reviewer to assess your manuscript. We are issuing a decision on your manuscript at this point to prevent further delays in the evaluation of your manuscript. Please be aware that the editor who handles your revised manuscript might find it necessary to invite additional reviewers to assess this work once the revised manuscript is submitted. However, we will aim to proceed on the basis of this single review if possible. 

We look forward to receiving your revised manuscript.

Kind regards,

Joanna Tindall, PhD

Staff Editor

PLOS ONE

Journal Requirements:

3. We note that your Data Availability Statement is currently as follows: All relevant data are within the manuscript and in Supporting Information files.

Reviewers' comments:

Reviewer's Responses to Questions

**Comments to the Author**

1. Is the manuscript technically sound, and do the data support the conclusions?

Reviewer #1: Yes

2. Has the statistical analysis been performed appropriately and rigorously? 

Reviewer #1: No

3. Have the authors made all data underlying the findings in their manuscript fully available?

Reviewer #1: Yes

4. Is the manuscript presented in an intelligible fashion and written in standard English?

Reviewer #1: No

5. Review Comments to the Author

Reviewer #1: There are several things that I recommend the authors to check:

Formula (4), as I understand from the formula, the range is no longer between 0 to 1 as in the initial cosine similarity.

In my understanding, rev1 and rev2 can be considered as weights. rev_1 is between 0 and 1, so it seems OK, but rev_2 is between e and 1; as these two are the multiplier of the initial cosine similarity, then the similarity is no longer between 0 and 1.

Moreover, if the item is popular, then Ni is large, then rev2 will to 1; but if the item is not popular then rev2 will close e

which will increase the similarity, i.e increase the sim (u.v), according to Formula (4). Isn't this contradictory?

It is stated following Formula (4) that the RHS is the classic cosine similarity. Well, I disagree with this, since it's the modified, since it is already weighed by rev1 and rev2

Formula (5): Ni(u) is the neighbors set of rating the same items with user u; does it mean they rate all the same items? Or it can be just some items are the same, and some others can be different? Or just the specified item-i? What if Ni(u) is an empty set? Moreover, please check the consequence of my concern about Formula (4) on the similarity in Formula (5).

Page 14: dimension K  what is K? how to determine K? is it related to the index k?

mu=average rating  how do you get the value? averaged over what?

|Test|  what is this? An explanation is required.

Page22: Formula (16) loss function: y.ycap  is it dot product? isn't y the rating, i.e scalar? Or is it a vector? Is it related to y_j= the feature items that user has purchased but not rated? If there is no value yet to y_j, then how can we do y.yhat?

Page 23. Formula (17) for sparsity, the part "(number of users)x(number of items)"  is it standard formula?

Otherwise, why is it not just 1- #rated items/#items? Since if using that formula, even if all the items are rated, there is still sparsity, that is 1-1/#users  this is kind of not really representing the idea of sparsity. Or is there other interpretation, or rationale of the sparsity in this context?

The number of nearest neighbors for users was set to 50  the rationale? as in kNN usually we use 5; but there might be specific consideration in this context of data? The concern is that the larger the number of neighbors used, the more heterogeneous the result will likely.

Page28: Formula (18), average of average? I think the formula is wrong, the 1st average is not needed.

Writing in general: it needs to be written in a standard way. Please pay attention to the punctuations, some spacing are not the same, there are paragraph contains only 1 or 2 sentences; there are improper sentences (e.g. sentence start with "And"), no space between words, typos e.g.: "analyss", "increasement, "Satisfcation". Moreover, please be consistent in the label of column names, some start with capital letter, some are not. Also, in captions of figures and tables, should be "self-explained" caption. In Fig.6 and Fig.7, there is different orientation for the y-axis label, please be consistent.

6. PLOS authors have the option to publish the peer review history of their article (what does this mean? ). If published, this will include your full peer review and any attached files.

**Do you want your identity to be public for this peer review?** For information about this choice, including consent withdrawal, please see our Privacy Policy .

Reviewer #1: **Yes: ** Sarini Abdullah

---

## [Author Response · Author response to Decision Letter 1]

22 May 2025

please refer to the attachment "response" for the detailed response content.

---

## [Decision Letter · Decision Letter 1]

1 Aug 2025

Bundle recommendation methods considering rating data differences for online retailers

PONE-D-24-56685R1

Dear Dr. Liu,

We’re pleased to inform you that your manuscript has been judged scientifically suitable for publication and will be formally accepted for publication once it meets all outstanding technical requirements.

Kind regards,

Carla Pegoraro

Staff Editor

PLOS ONE

Additional Editor Comments (optional):

Reviewers' comments:

Reviewer's Responses to Questions

**Comments to the Author**

1. If the authors have adequately addressed your comments raised in a previous round of review and you feel that this manuscript is now acceptable for publication, you may indicate that here to bypass the “Comments to the Author” section, enter your conflict of interest statement in the “Confidential to Editor” section, and submit your "Accept" recommendation.

Reviewer #2: All comments have been addressed

2. Is the manuscript technically sound, and do the data support the conclusions?

Reviewer #2: Yes

3. Has the statistical analysis been performed appropriately and rigorously? 

Reviewer #2: Yes

4. Have the authors made all data underlying the findings in their manuscript fully available?

Reviewer #2: Yes

5. Is the manuscript presented in an intelligible fashion and written in standard English?

Reviewer #2: Yes

6. Review Comments to the Author

Reviewer #2: I have no comments. the authors performed all the required revisions. it is accepted all comments are addressed

7. PLOS authors have the option to publish the peer review history of their article (what does this mean? ). If published, this will include your full peer review and any attached files.

**Do you want your identity to be public for this peer review?** For information about this choice, including consent withdrawal, please see our Privacy Policy .

Reviewer #2: **Yes: ** farah tawfiq abdul hussien

---

## [Editor Report · Acceptance letter]

PONE-D-24-56685R1

PLOS ONE

Dear Dr. Liu,

I'm pleased to inform you that your manuscript has been deemed suitable for publication in PLOS ONE. Congratulations! Your manuscript is now being handed over to our production team.

Kind regards,

on behalf of

Dr Carla Pegoraro

Staff Editor

PLOS ONE